# Intra-vector infection dynamics challenge how to model the extrinsic incubation period for major arboviruses: dengue, Zika, and chikungunya

**Léa Loisel**[1]*, **Vincent Raquin**[2], **Maxime Ratinier**[2], **Pauline Ezanno**[1☯],
**Gaël Beaunée**[1☯]

**1** Oniris, INRAE, BIOEPAR, Nantes, France, **2** EPHE, Université PSL, INRAE, Université Claude Bernard Lyon 1, IVPC, Lyon, France

☯ These authors contributed equally to this work.
* lea.loisel@inrae.fr

## Abstract

Arboviruses represent a significant threat to the health of humans, animals, and plants worldwide. Mechanistic modeling has proven useful for elucidating the transmission, anticipating the spread, and predicting the response of arboviruses to control measures. However, most models approximate the intra-vector infection dynamic (IVD), which occurs during the extrinsic incubation period (EIP), by a single stage with an average duration. At the end of this stage, all exposed vectors are expected to become infectious. Strong assumption is hidden behind this: that the EIP is exponentially distributed in the vector population. To assess the validity of this assumption, we developed a stochastic compartmental model that represents successive IVD stages, associated with the crossing or not of the three within-mosquito barriers (infection, dissemination, and transmission). We calibrated the model using an ABC-SMC (Approximate Bayesian Computation - Sequential Monte Carlo) method, which includes model selection. We searched for literature data on experimental infections of *Aedes* mosquitoes infected by dengue, chikungunya, or Zika viruses. We demonstrated the large discrepancy between the exponential hypothesis and observed EIP distributions for dengue and Zika viruses, and identified more relevant EIP distributions. This work provides a generic modeling framework that can be applied to other arboviruses for which similar data are available. Our model also can be linked to population-scale models to aid future arbovirus control efforts.

## Author summary

To tackle the worldwide threat posed by arboviruses like dengue, chikungunya, and Zika, we need accurate and reliable epidemiological models to better understand transmission dynamics, rank possible interventions, and provide timely

**Data availability statement:** Code and data used for inference have been deposited on GitHub at https://forge.inrae.fr/dynamo/ivd-modelling-and-inference.

**Funding:** This work was supported by INRAE Metaprogramme DIGIT-BIO (Digital biology to explore and predict living organisms in their environment), through the MIDIIVEC project to LL, VR, MR PE, GB. Funded by the European Union (WiLiMan-ID, grant agreement 101083833). Views and opinions expressed are however those of the author(s) only and do not necessarily reflect those of the European Union or the granting authority. Neither the European Union nor the granting authority can be held responsible for them. LL, PE and GB were funded by the WiLiMan-ID project. The funders had no role in study design, data collection and analysis, decision to publish, or preparation of the manuscript.

**Competing interests:** The authors have declared that no competing interests exist.

support to policymakers. However, our study clearly demonstrates that a key assumption often made by these vector-borne transmission models does not hold in most circumstances. Using data from infection experiments in the literature, our generic modeling framework of intra-mosquito infection dynamics enabled us to identify the most relevant distribution for the extrinsic incubation period. The extrinsic incubation period corresponds to the time required for a vector that has acquired a virus during a blood meal on an infectious host to become infectious (i.e., able to transmit the virus to a new host). This period is, therefore, a key process in vector-host transmission dynamics. This framework can be integrated into population-scale models to better characterize the mosquito extrinsic incubation period and improve model predictions.

## Introduction

Arboviruses, which are viruses transmitted to vertebrate hosts through the bites of arthropod vectors, were responsible for a third of emerging infectious disease events over the past two decades [1]. The prevalence of arboviruses has increased significantly worldwide [2], especially due to the intensified movement of people and goods, as well as environmental changes [3–5]. This expansion in the worldwide distribution of arboviruses, combined with the growing resistance of mosquitoes to insecticides and a lack of vaccines [6], complicates the control of these diseases [3,4]. Among the most important arboviral diseases affecting public health are those caused by the Zika (ZIKV), dengue (DENV), and chikungunya (CHIKV) viruses [7]. These viruses, mainly transmitted to humans by *Aedes* mosquitoes, are currently distributed over both hemispheres of the globe, making them a worldwide threat [8]. Improving our knowledge of their transmission dynamics to anticipate and limit future epidemics is one of the objectives proposed by the World Health Organization in its 'Global Initiative on Arboviruses' [9].

Arbovirus transmission is a complex process involving multiple stages, from the viral infection of the vector to the spread of the virus within host populations, that is influenced by biotic and abiotic factors [10]. Epidemiological models of transmission between vectors and hosts have been developed to better understand and forecast disease dynamics [11]. The accuracy and reliability of these models are essential to better guide and assist arbovirus surveillance and to help implement effective disease management interventions [12]. A key assumption of these models is how the extrinsic incubation period (EIP), which is a component of vector competence, is represented [13,14]. The EIP corresponds to the time required for a vector that has acquired a virus during a blood meal on an infectious host to become infectious (*i.e.*, able to transmit the virus to a new host) [15]. The transmission of a virus by a mosquito thus varies depending on the relationship between the EIP, the mosquito lifespan, and the biting rate. During the EIP, a virus must cross three barriers within the vector before it can reach the vector's saliva and become transmissible (S1 Fig). The intra-vector infection dynamic (hereafter referred to as IVD) corresponds to the

dynamic of the virus crossing these three barriers: (i) the infection barrier, crossed when the virus enters the vector intestinal epithelium; (ii) the dissemination barrier, crossed when the virus exits from the vector's midgut to reach the circulatory system, then spreads throughout the vector's body until it reaches the salivary glands, and finally the (iii) transmission barrier, crossed when the virus is excreted in the mosquito's saliva [16]. At this final stage, the virus can potentially be transmitted to a susceptible host during a subsequent blood meal.

IVD has been studied by experimental infection assays for various mosquito-arbovirus pairs [17,18], but this dynamic is inadequately addressed in current mechanistic models of vector-borne virus transmission. Recent studies combining experimental data and statistical modeling have shown that for DENV, variations in IVD influence both the size of epidemics and the probability of their onset [19]. Most compartmental transmission models do not explicitly represent IVD; instead, they provide an approximation with a single compartment whose entry rate depends on infection, dissemination, and transmission rates, and whose exit rate is assumed to be constant and equal to the inverse of the mean EIP [12,20]. While it is parsimonious, this mathematical formalism relies on a strong assumption: that the duration of the EIP follows an exponential distribution within the mosquito population. Yet for measles, for example, research has shown that the duration of the incubation period in hosts was better represented by a normal [21] or a gamma distribution [22]. In vector-borne diseases, the selected EIP distribution can impact the estimated virus transmission between vectors and hosts. This impact may take the form of strong variations in the basic reproductive ratio, as shown for bluetongue [23], or by shaping the intensity and timing of the epidemic peak in hosts [12] and the early increase in infected hosts [24], as shown for dengue. In most cases, modelers correctly assume that variations in EIP distribution will not significantly impact model predictions, although this has not been formally addressed. In light of potentially important inter-individual EIP variation among individual mosquitoes, it is worth accurately representing EIP distribution in mosquito populations, and considering its impact on pathogen transmission efficacy.

Using a combination of laboratory experiments and statistical models, Lequime et al. [25] explored how biotic and abiotic factors influence the IVD of ZIKV. Their findings underscore the need to develop a generic mechanistic model that extends this knowledge to other arboviruses. Recent studies have investigated the role of infectious dose in two of the three stages of IVD, either infection and dissemination, or infection and transmission, for DENV [26] and ZIKV [27,28]. For DENV, the viral titer has been explicitly represented, while for ZIKV, both the viral titer and the number of infected cells have been modeled. These contributions have significantly advanced our understanding of the biotic factors affecting barrier crossing in infected mosquitoes. However, a generic mechanistic model of mosquito stages during IVD is still lacking. Such a model would make it possible to simultaneously estimate the proportion of mosquitoes in which viruses fail to cross within-mosquito barriers and the duration distribution of IVD stages. Typically, these parameters are estimated independently, making it impossible to account for their potential correlations in transmission models.

We aimed to better characterize the IVD for DENV, ZIKV, and CHIKV, three arboviruses of major public health importance. To evaluate the validity of assuming an exponentially distributed EIP, we developed a stochastic mechanistic model of IVD. This model explicitly represents the distribution of IVD stage durations and the crossing of each of the three within-mosquito barriers. To examine how these processes vary according to biotic factors, such as virus species or strains, mosquito species, and infectious doses, we calibrated the model using a comprehensive dataset of published experimental results obtained under comparable protocols.

Our analysis simultaneously estimated the proportions of mosquitoes in which viruses fail to cross each of the three barriers and the duration distributions in the infection and dissemination stages. The results revealed significant deviations from the exponential assumption, underscoring the importance of using more realistic distributions of IVD stage durations. Moreover, our model generated consistent parameter sets that will enable the explicit incorporation of experimental results into more comprehensive vector-host transmission models. This approach facilitates the representation and exploration of links between intra-vector processes and the larger-scale dynamics of arbovirus transmission. By enhancing the robustness of existing epidemiological models, this new knowledge could contribute to improved epidemic forecasting and more effective public health interventions for mosquito-borne diseases.

## Results

### The EIPs of DENV and ZIKV are not exponentially distributed

Using vector competence data available in the literature for DENV, ZIKV, and CHIKV, we inferred the parameters of our IVD model (Fig 1). The model includes one compartment for each IVD stage. Stage *E* represents exposed mosquitoes, when the virus is present in the digestive tract following a blood meal. Stage *I* corresponds to infected mosquitoes, when the virus is present in the midgut. Stage *D* denotes disseminated mosquitoes, when the virus has spread to the circulatory system. Finally, stage *T* refers to transmitter or infectious mosquitoes, marked by the presence of the virus in the saliva. To account for the non-systematic crossing of barriers [16], we introduced three parameters, $\gamma_I$, $\gamma_D$, and $\gamma_T$, to represent the proportions of mosquitoes for which the infection, dissemination, and transmission barriers are crossed, respectively. When crossing the first or second barrier and entering in IVD stage *I* or *D*, mosquitoes were randomly distributed among the sub-compartments of this new stage using a multinomial distribution to represent the duration distribution in the given stage. This distribution can be mediated either by an exponential distribution (of parameter $\lambda$) or by a beta distribution (of parameters $\alpha$ and $\beta$), depending on the model used. The models were called IbetaDexpo, IbetaDbeta, IexpoDexpo, and IexpoDexpo, and they differ based on the distribution used - either beta or exponential (expo) - for the infected (*I*) and disseminated (*D*) stages, as indicated by the first and second parts of their names, respectively. To perform inference, we used an Approximate Bayesian Computation - Sequential Monte Carlo (ABC-SMC) approach combined with model selection. We studied 49 scenarios, each consisting of a vector competence experiment on *Aedes* female mosquitoes from a specific species and geographical origin. These mosquitoes were exposed to a virus isolate from a given species at a single infectious dose (expressed as $\log_{10}$ (FFU/mL)) in the blood meal (Fig B and Tables A, B, C and D in S1 Appendix).

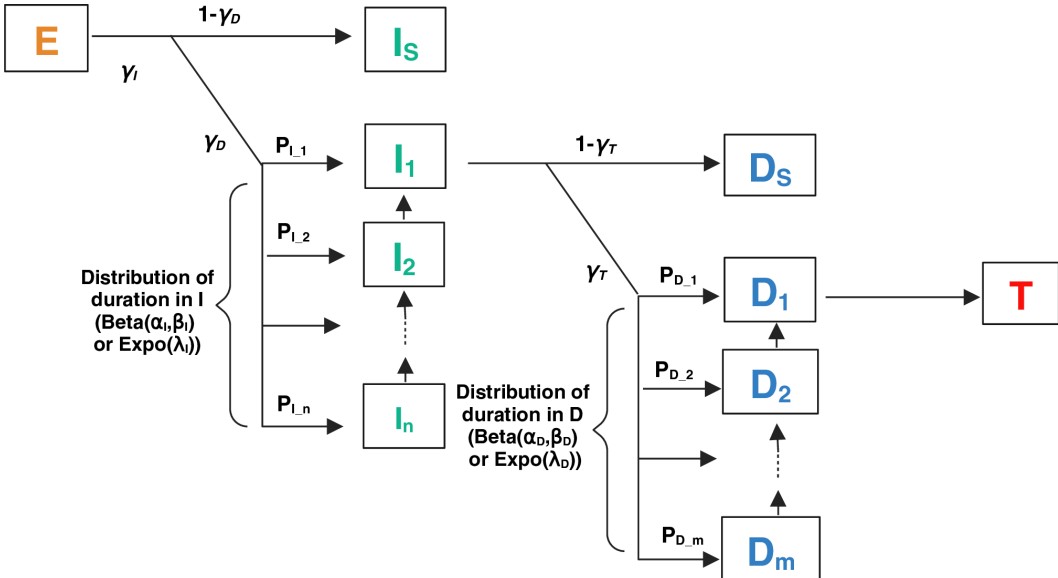

**Fig 1. Conceptual diagram of the intra-vector infection dynamic model.** Each model compartment represents a vector stage (*E*: exposed vector, *i.e.*, a vector with a virus in its digestive system; *I*: infected vector, *i.e.*, a vector with a virus in its midgut epithelium, with $I_S$: infected vector remaining in *I*, $I_1$ to $I_n$: infected vector remaining 1 to *n* days in *I*; *D*: disseminated vector, *i.e.*, a vector with a virus in its circulatory system, with $D_S$: infected vector remaining in *D*, $D_1$ to $D_m$: infected vector remaining 1 to *m* days in *D*; *T*: infectious vector, *i.e.*, a transmitter vector with a virus in its saliva. Model parameters are: $\gamma_I$, $\gamma_D$, $\gamma_T$, the proportions of mosquitoes for which the infection, dissemination, and transmission barriers are crossed, respectively; $\alpha$ and $\beta$, the beta law parameters; $\lambda$, the exponential law parameter; *n* and *m*, the maximum length of stay in *I* and *D*, respectively; $P_{I\_1}$ to $P_{I\_n}$, the probabilities for mosquitoes to be distributed respectively in the $I_1$ to $I_n$ compartments; $P_{D\_1}$ to $P_{D\_m}$, the probabilities for mosquitoes to be distributed respectively in the $D_1$ to $D_m$ compartments. Created in BioRender. https://BioRender.com/c67z722.

Depending on the experimental design, we inferred either a partial model (EID), when no information was available for the transmitter stage, or a complete model (EIDT), when the data set covered all IVD stages. As the parameters of our IVD model were estimated together, our study provides a coherent set of parameters for the experimental conditions that we examined.

We have shown that it is often unrealistic to assume that a large proportion of mosquitoes become immediately infectious after taking an infectious blood meal, as presumed by the exponential distribution (Fig 2). For the infected stage, the main distribution selected (*i.e.*, the one most frequently selected among those providing the best fit to the data) was the beta distribution for 7 out of 7 scenarios tested for DENV (Fig 2Ai) and for 10 out of 15 scenarios tested for ZIKV (Fig 2Bi). For the disseminated stage, the main distribution selected was again the beta distribution for 6 out of 7 scenarios tested for DENV (Fig 2Ai) and 10 out of 15 scenarios tested for ZIKV (Fig 2Bi). Conversely, an exponential distribution was more often selected for CHIKV for both stages (6/10 and 10/10 scenarios for the infected and the disseminated stages, respectively (Fig 2Ci)). We obtained similar results when inferring the parameters of the partial EID model (Fig A in S2 Appendix).

To assess the precision of the estimated parameters governing the distributions in the infected and disseminated stages, we focused on the dispersion of the selected density distributions instead of the individual parameter values. Indeed, two very different parameter sets can yield similar distribution shapes. For the exponential distributions, we observed a near-complete visual overlap of almost all selected density distributions for each scenario, indicating a low level of uncertainty (Figs C-AY in S2 Appendix). In contrast, the visual assessment of beta distributions was more challenging. Therefore, we quantified the dispersion among all of the beta distributions selected for each scenario by applying a Kolmogorov-Smirnov test, and we calculated the percentage of statistically similar beta distributions (p-value > 0.05). In the infected stage, 13 out of 21 scenarios using the EIDT model, and 11 out of 15 scenarios using the EID model (Table K in S3 Appendix), showed more or at least 50% similarity among their beta distributions. Conversely, precision appeared lower for the disseminated stage, with only 7 out of 16 scenarios exhibiting more or at least 50% statistically similar beta distributions (Table L in S3 Appendix).

The values inferred for the barrier parameters indicated that there was a non-systematic crossing of the three barriers (Fig 3). However, the probabilities of crossing the infection ($\gamma_I$) and dissemination ($\gamma_D$) barriers were statistically above 0.9 in one-third to one-half of the scenarios (using a Wilcoxon test, for 10/32 and 9/32 scenarios, respectively, with the EIDT model (Fig 3A and 3B; Tables A and C in S3 Appendix); and for 10/17 and 8/17 scenarios, respectively, with the EID model (Fig B in S2 Appendix; Tables B and D in S3 Appendix)). These results demonstrate that the infection and dissemination barriers are often weak. The probability of crossing the transmission barrier ($\gamma_T$) was statistically less than 0.5 in two-thirds of the scenarios (using a Wilcoxon test, for 23/32 scenarios with the EIDT model (Fig 3C; Tables A and D in S3 Appendix)). This emphasizes the key role of this last barrier during IVD [29].

For the barrier-crossing parameters ($\gamma_I$, $\gamma_D$, $\gamma_T$), at least one-half of the scenarios studied had a 90% credibility interval (CI) < 0.15 (Figs 3 and B in S2 Appendix). Specifically, the 90% CI for $\gamma_I$ were < 0.15 for 21 out of 32 scenarios with the EIDT model (Table A in S3 Appendix), and 14 out of 17 scenarios with the EID model (Table B in S3 Appendix). In contrast, the 90% CI were wider for $\gamma_D$, with only 10 out of 32 scenarios (EIDT) and 7 out of 17 scenarios (EID) showing intervals < 0.15. The uncertainty was even greater for $\gamma_T$, with 90% CI < 0.15 observed in just 4 out of 32 scenarios with the EIDT model (Table A in S3 Appendix). The uncertainty in inference results was influenced by the size of mosquito samples and increased for scenarios where $\gamma_I$ was low. Specifically, except for two cases, all scenarios with $\gamma_I$ below 0.5 had 90% CI for $\gamma_D$ and $\gamma_T$ > 0.4 (Table A in S3 Appendix).

## Model fit quality raises questions about IVD and its underlying assumptions

The quality of the model inference was evaluated based on the visual fit between observed and simulated data for the selected models (Figs 2A–2Ciii and A and Fig C-AY in S2 Appendix). Specifically, 25 out of 32 scenarios showed a good

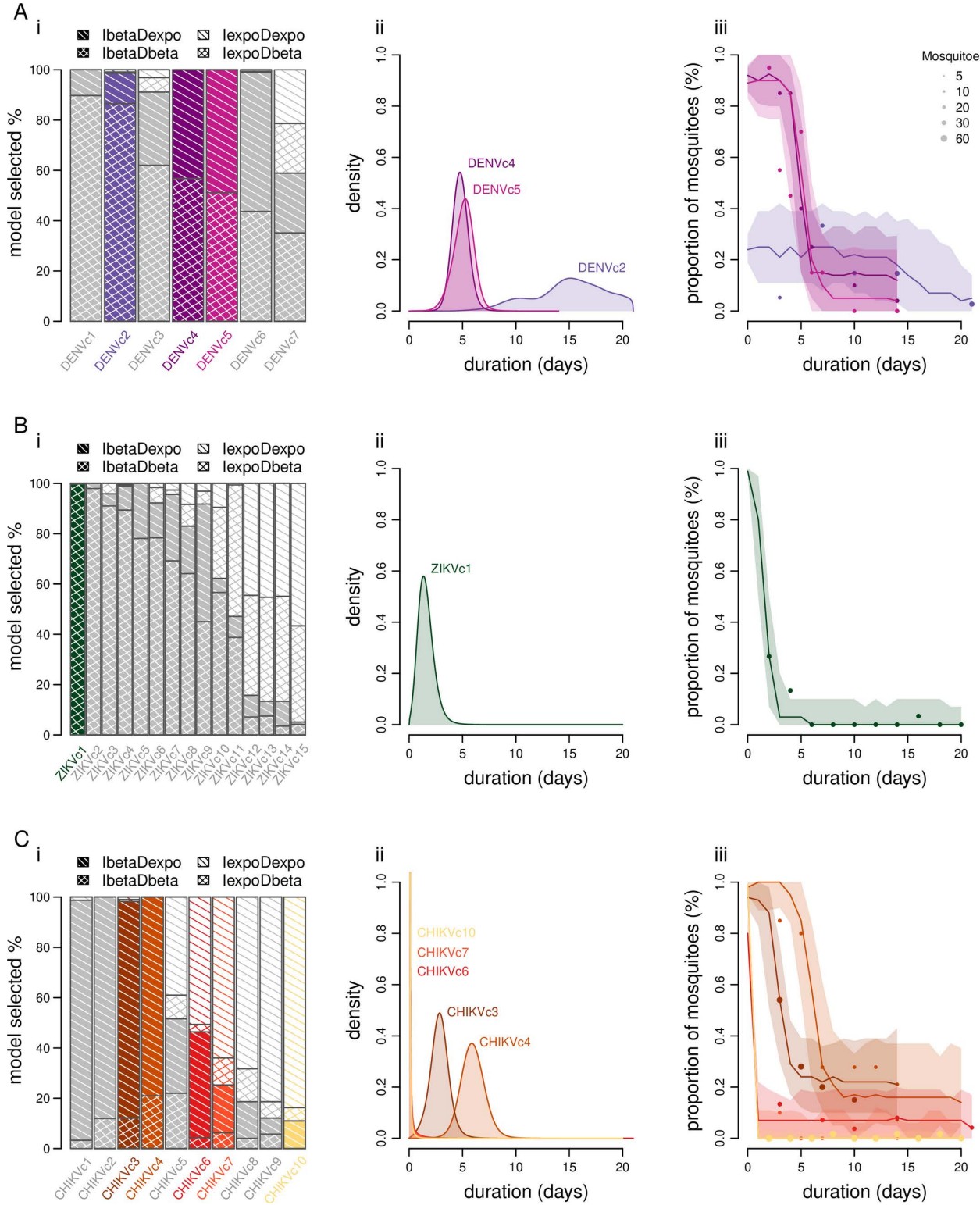

**Fig 2. Inference of the duration distributions of the intra-vector infection dynamic stages for each scenario.** One scenario consists of a vector competence experiment on female mosquitoes of a specific genus, species, and geographical origin, exposed to a virus isolate of a given species at a single infectious dose ($\log_{10}$ FFU/mL) in the blood meal. A: dengue virus (DENV); B: Zika virus (ZIKV); C: chikungunya virus (CHIKV). (i): Proportion of selected models among those tested: IbetaDexpo, IbetaDbeta, IexpoDexpo, IexpoDbeta (these models differ based on the distribution used - either

beta or exponential - for the infected and disseminated stages, as indicated by the first and second parts of their names, respectively). Scenarios with 5 or more days post-exposure (Dpe) are shown in color (one color per scenario), others are in gray. (ii): Average of the selected distributions in the infected stage for the main selected model (only scenarios with 5 or more Dpe are represented). (iii): Selected dynamics in the infected stage for the main selected model (only scenarios with 5 or more Dpe are represented). Dots: observed data; lines: mean dynamics; uncertainty ribbons (5%-95%): selected simulated dynamics.

visual fit for the three IVD stages using the complete EIDT model (Table F in S3 Appendix). Similarly, all 17 scenarios studied using the partial EID model showed a good visual fit for the first two IVD stages (Table G in S3 Appendix). This performance was further supported by the mean of the root mean squared error (RMSE): 21/32, 23/32 and 26/32 scenarios had a mean RMSE below 5 respectively for the infected, disseminated, and transmitter stage using the EIDT model (Table H in S3 Appendix). For the EID model, all scenarios had a mean RMSE below 5 for the infected and disseminated stages (Table I in S3 Appendix).

The lower inference quality for some scenarios nevertheless raises questions about the processes assumed in IVD. Indeed, the quality of fit decreased for scenarios where the dynamics observed in the infected, disseminated, and transmitter stages deviated from the expected IVD. Classically, entomologists assume that mosquitoes become infected within a few hours [30,31] and then progress unidirectionally through stages with no possibility of returning [32]. As a result, we expect a decrease in the proportion of mosquitoes in the infected stage over time, an increase and then a decrease in the disseminated stage, and finally a continuous increase in the transmitter stage. However, in nine scenarios (DENVc1, DENVc2, ZIKVc4, ZIKVc5, ZIKVc6, ZIKVc10, ZIKVc12, ZIKVc13, ZIKVc15), the proportion of mosquitoes in the infected stage initially increased, and for eleven scenarios (CHIKVc1, CHIKVc2, CHIKVc3, CHIKVc6, CHIKVc7, CHIKVc8, CHIKVc9, DENVc6, DENVc7, ZIKVc1, ZIKVc11), the proportion of mosquitoes in the transmitter stage first increased but then decreased (Table J in S3 Appendix). These discrepancies probably reflect biological processes that were not considered in our model.

## Discussion

Our study, which focuses on DENV, CHIKV and ZIKV - three major human arboviruses - took into account all open access published vector competence experiments (that we could find) with comparable protocols. This provides enough material to draw robust conclusions for the three arboviruses studied and to estimate simultaneously all of the parameters of the intra-vector infection dynamic (IVD). We highlighted that the duration distributions of the first two IVD stages (midgut infection and dissemination in the circulatory system) are often non-exponential under the experimental conditions considered. We also noted that the viruses cross each of the three within-mosquito barriers in only a small number of exposed mosquitoes, with the transmission barrier posing the primary obstacle.

First, we concluded that it is often unrealistic to assume that the duration of IVD stages, and therefore of the extrinsic incubation period (EIP), follows an exponential distribution. This is particularly true for DENV and ZIKV, for which at least one of the two main distributions selected in the infected or disseminated stage was not exponential in over 80% of the experimental conditions studied. This exponential distribution assumption has already been called into question [20,33]. A meta-analysis of vector competence experiments for DENV has shown that a log-normal distribution was more appropriate for the EIP [34]. Our generic and mechanistic modeling framework makes it possible to select the most relevant EIP distribution for any experimental conditions and associated data, without the need to aggregate information across experiments. Our model, which was used here for three major arboviruses, could easily be used for other arboviruses such as the West Nile virus, provided that similar experimental data are available.

Second, our model enables us to represent and quantify the probabilities of viruses crossing each of the within-mosquito barriers according to experimental conditions. We highlighted that the transmission barrier was the most challenging to cross regardless of the mosquitoes, viruses, or infectious doses being considered [29]. Most vector-borne disease transmission models already take such barriers into account by using a host-to-mosquito transmission rate. However, the value

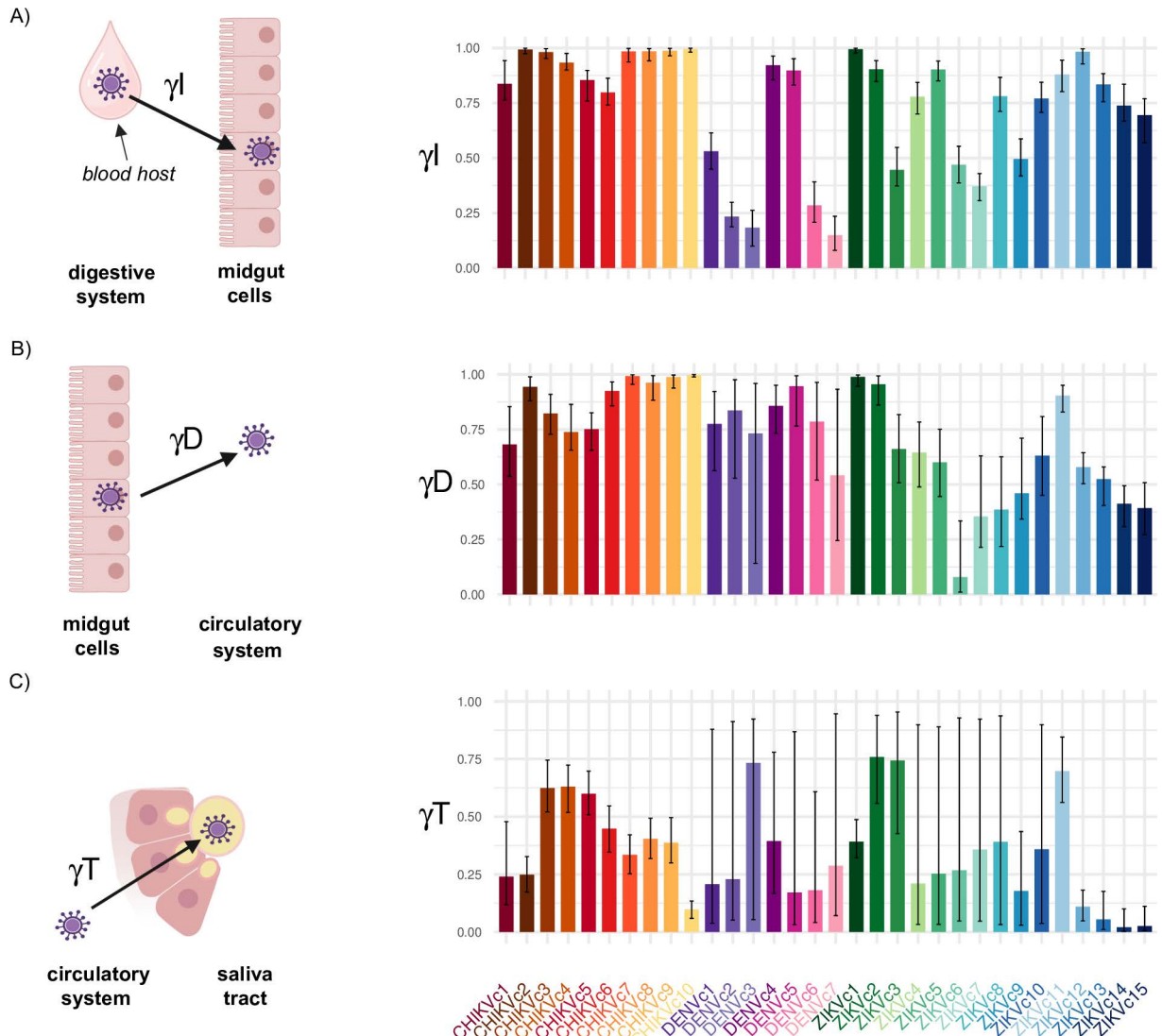

**Fig 3. Parameter estimates for the probabilities of crossing the infection, dissemination, and transmission barriers for each tested scenario\* for chikungunya virus (CHIKV), dengue virus (DENV), and Zika virus (ZIKV).** A: Infection barrier; B: Dissemination barrier; C: Transmission barrier. Each bar represents the parameter mode for a given scenario, with one color per scenario, and with the 90% credibility interval represented by the error bar. \**One scenario consists of a vector competence experiment on* Aedes *female mosquitoes from a specific species and geographical origin, exposed to a virus isolate from a given species at a single infectious dose in the blood meal*. Created in BioRender. https://BioRender.com/b77v427.

of this rate is usually chosen independently of the EIP duration (*e.g.*, with values taken from different sources), even though it is probable that these two parameters are correlated. Our modeling framework offers the possibility to estimate the three barrier-crossing parameters directly from experimental data, and together with the duration distributions of the infected and disseminated stages, thereby providing a consistent set of parameters and consolidated information.

A major contribution of our model is that it allows for the simultaneous relaxation of the exponential assumption while explicitly representing the intra-mosquito barriers when the model is used as a building block in larger population-scale epidemiological models (Fig C in S1 Appendix). It therefore proposes a new way to represent exposed vectors in population models. As our inference provides probability distributions for IVD parameters rather than single point estimates,

our study offers a framework to capture the inherent variability in IVD. This variability may reflect differences in vector competence between mosquito populations [35], as well as the genetic diversity and evolution of the viral population during incubation. Indeed, each intra-mosquito barrier creates a bottleneck that changes the viral population and shapes genetic diversity [36]. Coupling our model with population models could help to assess how IVD variability could influence the transmission dynamics of vector-borne diseases [12,23,24]. However, incorporating our framework into such models would require a new formulation of $R_0$, as standard expressions would no longer be applicable due to the assumption of an exponential EIP distribution. Finally, as our model explicitly represents the different IVD stages, it could be used to study these stages and barriers separately. This approach could support a control strategy based on reducing vector competence through genetic modification that targets one of these stages [37].

Our approach also could help establish a functional link between IVD and biotic factors (mosquito and virus genotypes, and viral infectious dose) to predict IVD outside the experimental conditions tested. Indeed, besides influencing the duration of the EIP [10,14], biotic factors also impact the distributions of residence times in IVD stages and the probabilities of crossing intra-mosquito barriers [10] in the experimental conditions considered. However, establishing a quantitative statistical link between biotic factors and IVD parameters requires significantly more experimental data than what is now available in the literature. While numerous studies provide experimental data on vector competence, there is a scarcity of data that enable estimations of the dynamic aspects of IVD. Specifically, there are currently very few studies available that track infected, disseminated, and transmitter stages across a sufficient number of time points (at least 4 in our study), with a sufficient number of mosquitoes per time point (at least 20), and that provide tables or raw data. Out of the 182 articles with vector competence data on CHIKV, ZIKV, and DENV analyzed in our study, only 14 met this criterion. Furthermore, we might have missed articles providing relevant data since the tools used for systematic searches of articles are not perfect. Nevertheless, the variety of experimental data that already exists - and which was used in our study – is sufficient to lead to robust conclusions, and enables us to propose a modeling framework that will be easy to use on new data once it becomes available. Establishing such a functional link between IVD and biotic factors would be essential in the future to account for the spatial and temporal variability of IVD in population models.

A crucial step in this direction is to increase access to and sharing of standardized raw experimental data. Alongside the need for additional laboratory work using similar experimental protocols, there is a need to improve the reusability of published experimental data, *e.g.*, in meta-analyses or to address new questions. The published data used here came from studies on vector competence, each using specific experimental conditions. After selecting the most relevant studies (Fig A in S1 Appendix)we were able to re-use these data to better understand IVD. However, a strong limitation lay in finding and properly using the studies. First, when we searched for articles and compared the ones we obtained with the ones included in a review using a very similar query [38], we observed many different records (Fig A in S1 Appendix), possibly due to varying terminology [38]. This highlights the difficulty of being exhaustive in a literature review due to the lack of standardization in the description of studies and in the terms used [38]. Furthermore, several of the articles retrieved did not meet the necessary criteria for inclusion, either because they were not openly accessible or because they consisted solely of figures, without tables or raw data. Sharing raw and standardized data as proposed by Wu *et al.* [39] and following the FAIR (Findable-Accessible-Interoperable-Reusable) principle [40] is crucial for increasing the value and usefulness of data [39]. One potential solution to the issue of data sharing is the establishment of an online platform [38]. An example of such a platform is the COMET platform (currently under construction) from the Verena program [41].

One limitation of our study is that the data used to infer model parameters came solely from laboratory experiments, which are sometimes far removed from field conditions [42]. The parameters governing IVD might be influenced by the type of blood feeding (artificial *versus* on an infected animal), as is the case for mosquito infection rates [43]. Here, we used data from experiments involving artificial blood feeding as there was insufficient suitable data from experiments performed with blood feeding on an infected vertebrate host. Such experiments are rare and challenging to conduct due

to evident ethical and logistical constraints [44,45]. However, in the rare instances where this type of data is available, our modeling approach can be used without needing to make any changes.

Experimental and modeling studies have reciprocal benefits. While models are fed knowledge from experimental (and field) observations, experimental designs can benefit from a modeling framework such as the one proposed here. First, our framework can help define the number of time steps to be observed, which are often governed by cost and time limitations. Second, our framework also enables the identification of the most relevant time points to observe for gaining as much information as possible. For CHIKV, we showed that the vector's EIP is often exponentially distributed, and that the associated IVD is generally very fast. As a result, access to several early time points would provide more valuable information for this disease than later time points. In contrast, for ZIKV and DENV, which are characterized by a much slower IVD and a vector EIP that is barely exponentially distributed, access to later time points would be important. Our framework can also help determine the most appropriate duration for vector competence experiments. For example, the shapes of the exponential and beta distributions inferred for the infected and disseminated stages indicate that most mosquitoes had already gone through these stages long before the experiments ended. This validates the model's assumption regarding these maximum durations and confirms the commonly used time windows for vector competence experiments. A second benefit relates to identifying the optimal sample size for mosquitoes. Indeed, this directly affects the uncertainty of the inference results, which our framework also quantifies. In half of the scenarios studied, the sample size used in the experiments was sufficient to obtain a low result uncertainty. However, when the barriers to infection or dissemination were high, the number of mosquitoes in the final stage was insufficient to infer parameters with a low uncertainty. Our approach could help determine in advance how many mosquitoes to sample at each time point to gather sufficient information, assuming we have a preliminary idea of the concerned IVD. For example, we suggest that researchers increase the number of mosquitoes sacrificed at intermediate and final time points of the experiment, as this is when the transmission barrier is often crossed. This is particularly important for virus/mosquito/dose conditions, for which we expect infection, dissemination or transmission barriers to be strong. This approach would ensure that there are enough mosquitoes to accurately study the duration distribution in the disseminated stage and the dynamics of crossing the transmission barrier.

Our model represents a foundational brick that may undergo future refinement to more accurately reflect the underlying processes involved in IVD. While the quality and uncertainty of the inferences were satisfactory for most scenarios, there were some for which simulations and observations did not align. This discrepancy is potentially due to certain mechanisms that have not yet been incorporated into the model.

Further investigations could focus on adding a stage between exposed and infected vectors to represent individuals infected but below the detection threshold. Indeed, in some of the experiments used here [46–48], the number of mosquitoes observed in the infected stage increased between the first and second time point studied. This has been observed in some other competence experiments [49–51], but our model does not allow such dynamics, as it assumes the infected barrier is crossed within the first few hours after the infected meal, consistent with the duration of digestion [30,31]. Consequently, the number of mosquitoes in the infected stage can only decrease. A potential explanation of the increase observed could lie in the latency period between the infection of epithelial cells and virus replication within these cells. This could lead to a delay in reaching the detection threshold for viruses in mosquito bodies [26,52]. On one hand, adding a stage to the model would facilitate a more comprehensive understanding of the dynamics during the infected stage at the start of the experiment. On the other hand, parameterizing the entry and exit from this new compartment appears quite challenging, especially due to the difficulty in experimentally distinguishing a mosquito whose gut is "in the process of infection" from a mosquito whose midgut is fully infected.

A second modification to better reflect the observed dynamics could involve allowing for an exit from the transmitter stage. This has not been included in our model given the prevailing consensus that, once infected, the virus is not eliminated from the mosquito's body [32]. Nevertheless, recent studies on CHIKV have highlighted a reduction in the proportion of mosquitoes with the virus present in their saliva at later time points, which challenges this common assumption

[13,53]. The authors explain that this rare phenomenon could be a false negative detection due to an experimental issue (such as no expectorated saliva or a problem with the detection threshold). However, they also mention the possibility that the virus could be eliminated by the mosquito's immune system or through the natural inactivation of the virions [13]. As our model could not capture these dynamics, there was a discrepancy between the observed and simulated data in a few scenarios for CHIKV. Modeling the exit from the transmitter stage would require more knowledge about this phenomenon.

Another modification could be to consider the effect of co-infections [54,55] or multiple blood meals on IVD, with the aim of developing a more accurate representation of the natural behavior of mosquitoes [56,57]. Recently, Armstrong *et al.* [33] demonstrated that two successive blood meals increased the dissemination of CHIKV, DENV, and ZIKV in *Aedes aegypti*. This suggests that neglecting blood meal recurrence in the IVD model might lead to a slight underestimation of the dynamics in the disseminated stage. Incorporating this process would lead to a change in the model type, making an individual-based model more appropriate. Furthermore, inferring this process would require experimental data that are not currently available.

The present study proposes a generic modeling framework that enables researchers studying the influence of varying IVD characteristics on vector-host transmission dynamics. It is applicable to other arboviruses and provides a building block to improve the predictive capacity and accuracy of population-scale models. Such transmission models are one of the tools used to identify efficient control measures to fight arboviruses. Improving their accuracy would help limit the significant threat that vector-borne diseases pose to public health in many parts of the world, a threat which is increasing due to climate change.

## Materials and methods

### Experimental data

To calibrate our model, we collected relevant vector competence experimental data by performing an extensive literature search (Fig A in S1 Appendix)). We used four search databases (WOS core collection, Scopus, Pubmed, and Medline) with the following query: ("arthropod-borne" OR arbovirus) AND (competence OR "vector competence" OR "intra-mosquito" OR "intra-vector" OR dissemination OR transmission OR "systemic infection" OR "extrinsic incubation period" OR dynamic) AND (virus) AND("in vivo" OR empirical OR experimental OR saliva OR "vector competence assays") NOT tick. This query yielded 2197 records. After deleting the duplicates, we examined the articles by title and deleted 815 of them. We then examined 239 articles based on their abstracts. We excluded articles dealing with arboviruses other than CHIKV, DENV or ZIKV or not including experiments on vector competence. We also chose to exclude articles that were not in open access because we wanted to encourage open science. Next, we screened the remaining 102 articles. We removed articles studying only one IVD stage, those with only figures and no accessible tables or raw data, those with less than 20 mosquitoes sampled by time point, and those with less than 4 observed time points to increase our chances of having a sufficient number of time points to capture intra-vector viral dynamics. We added other articles from a review with a close query [38], and articles from our personal research helped with two reviews [58,59]. Our entire search process led to a total of 17 articles [13,19,25,29,46,47,53,60–69] (Tables A, B, C and D in S1 Appendix) encompassing 49 scenarios. Each scenario consists of a vector competence experiment on *Aedes* female mosquitoes from a specific species and geographical origin, exposed to a virus isolate from a given species at a single infectious dose (expressed as $\log_{10}$ FFU/mL) in the blood meal. The articles selected used closely similar experimental protocols (Fig B in S1 Appendix) and methods for identifying the IVD stages to which mosquitoes belong. The fully engorged females were kept under consistent conditions of temperature (°c), humidity (%), and light. They were sacrificed per group of 20–60 individuals on specific days post exposure (Dpe). A minimum of 4 observed time points was required for experiments to be included in the present study (maximum number of observed time points = 10). To detect the virus, different titration methods were used (reverse transcription polymerase chain reaction (RT-PCR) assay, focus-forming assay (FFA), plaque forming assay (PFA)). The presence of the virus was searched for in

different parts of the mosquitoes' bodies (saliva, thorax, abdomen, legs, wings, head) to determine the stage of each individual mosquito (i.e., infected, disseminated, or transmitter) at each Dpe. For two articles [29,69], we used only part of the experimental data, as they differed from other experiments in that they used a genetic mutant of CHIKV [29] or mosquitoes exposed to an insecticide over several generations prior to the gorging phase [69].

## Model design

To represent the barriers and IVD stages, and to assess the distribution of residence times in the infected and disseminated stages, we developed a discrete-time stochastic mechanistic compartmental model with a one-day time step (Fig 1). This model represents the same processes observed in the vector competence experiments (Fig B in S1 Appendix). The model includes one compartment for each IVD stage. Stage $E$ represents exposed mosquitoes, when the virus is located in the digestive tract following a blood meal. Stage $I$ corresponds to infected mosquitoes, when the virus is present in the midgut. Stage $D$ denotes disseminated mosquitoes, when the virus has spread to the circulatory system. Finally, stage $T$ refers to transmitter or infectious mosquitoes, marked by the presence of the virus in the saliva. To account for the non-systematic barrier crossing [16], we introduced three parameters, $\gamma_I$, $\gamma_D$, and $\gamma_T$, to represent the proportions of mosquitoes for which the infection, dissemination, and transmission barriers were crossed, respectively. We assumed that once a barrier was crossed, there was no way back, thus the virus would not be eliminated from the mosquito's body [32]. For mosquitoes in which the virus did not cross one or two barriers, we added two compartments $I_s$ and $D_s$ ('s' for stop), denoting mosquitoes in which the infection barrier but not the dissemination barrier, and the dissemination barrier but not the transmission barrier, had been crossed, respectively. Mosquitoes in which the virus had not crossed the infection barrier stayed in the $E$ compartment.

To model the distributions of IVD stage durations, we divided the infection and dissemination compartments into $n$ and $m$ sub-compartments, with $n$ the maximum duration in $I$ in days, and $m$ this duration minus one day. The maximum duration in $I$ was here set at the experiment duration. The maximum duration in $D$ was set at the experiment duration minus one day. These durations were chosen because the durations of the experiments, which ranged from 12 to 28 days, were close to the lifespan of mosquitoes [70]. This approach ensured that the experiment duration encompassed the mosquito biting period. In addition, a maximum duration in $I$ or $D$ longer than the experiment could not be calibrated with the experimental data.

When crossing the first or second barrier and entering in IVD stage $I$ or $D$, mosquitoes were randomly distributed among the sub-compartments of this new stage using a multinomial distribution to represent the duration distribution in the given stage. This distribution can be either mediated by an exponential distribution (of parameter $\lambda$) or by a beta distribution (of parameters $\alpha$ and $\beta$). The beta distribution allowed us to obtain a wide range of shapes depending on $\alpha$ and $\beta$ values, without the need for prior knowledge. As this distribution was defined on the interval [0,1], it was well suited to represent proportions. We bounded $\alpha$ and $\beta$ between 1 and 100 to avoid distributions tending toward infinity in 0 and 1.

Due to mortality, the number of mosquitoes sampled varied across observation dates. Thus, we introduced parameter $N_{tot}$, representing the number of mosquitoes sampled at each observation date, adjusted for each scenario and observation date to match experimental data. This parameter was integrated into the data-driven observation layer of the model to provide the number of mosquitoes per compartment per observation date. The model was implemented in R. Each model stochastic replicate $k$ with parameter set $(\gamma_I, \gamma_D, \gamma_T, (\alpha_I, \beta_I)$ $or$ $\lambda_I, (\alpha_D, \beta_D)$ $or$ $\lambda_D)_k$ represented one vector competence experiment and provided outputs comparable to those obtained in the laboratory experiments, *i.e.*, mosquito numbers in infected, disseminated, and transmitter stages for each observed Dpe.

## Model calibration

We inferred model parameters using an Approximate Bayesian Computation (ABC) approach, with a Sequential Monte Carlo (SMC) sampler [71]. This iterative algorithm improves the basic ABC algorithm by incorporating two main steps: weighted resampling of simulated particles, and a gradual reduction in tolerance. As in the ABC rejection approach, a

prior distribution is defined, aiming to estimate a posterior distribution. However, in ABC-SMC, this estimation is achieved sequentially by constructing intermediate distributions in each iteration, converging toward the posterior distribution. Our specific implementation of the algorithm (R package BRREWABC) improves upon Del Moral et al.'s (2006) original algorithm [71] in three ways: (i) an adaptive threshold schedule selection based on quantiles of distances between simulated and observed data [72,73]; (ii) an adaptive perturbation kernel width during the sampling step, dependent on the previous intermediate posterior distribution [74,75]; and (iii) the capability to use multiple criteria simultaneously. We added a step to the inference process to select the best model [76] among four options: IbetaDbeta, IbetaDexpo, IexpoDbeta, and IexpoDexpo. These models differ based on the distribution used - either beta or exponential - for the infected and disseminated stages, as indicated by the first and second parts of their names, respectively. Depending on the experimental design, we inferred either a partial model (EID), when no information was available for the transmitter stage, or a complete model (EIDT), when the data set covered all IVD stages. This process resulted in the inference of 4 or 7 parameters, respectively: three proportions ($\gamma_I$, $\gamma_D$, $\gamma_T$) and distribution law parameters for $I$ (($\alpha_I$, $\beta_I$) or $\lambda_I$) and $D$ (($\alpha_D$, $\beta_D$) or $\lambda_D$) stages. The ranges of variation of each of inferred parameters and their justifications are presented in Table 1. The summary statistics used corresponded to the number of mosquitoes in each stage at each Dpe. For each particle, we calculated four distances, one for each of the four IVD stages, defined as the sum of the squared errors between the observed and simulated data for the different Dpe. A particle was accepted if each of the four associated distances met its own acceptance criteria, *i.e.*, was below the respective threshold. The thresholds for these criteria were recalculated at each inference round, using the 90th quantile of particles accepted in the previous round. In each round, 800 particles were accepted.

## Statistical analysis

To analyze the values of $\gamma_I$, $\gamma_D$, and $\gamma_T$, a Wilcoxon test was performed using R to statistically compare these parameters to 0.9 in order to assess the high values, and to 0.5 in order to assess the low values [77].

We first visually assessed the fit quality between observed and simulated data. Then, we calculated the mean of the root mean squared error (RMSE) for each dynamic in the infected, disseminated, and transmitter stages, for all simulation results and each scenario, using the simulated and observed mosquito numbers at each Dpe. The mean RMSE was expressed in mosquito numbers, with a mean RMSE lower than 5 indicating a good fit.

To study the uncertainty of parameter values, we used one of three methods, depending on the parameter concerned. For the crossing probabilities ($\gamma_I$, $\gamma_D$, $\gamma_T$), we evaluated the degree of confidence in the 90% credibility interval (90% CI) by comparing its width to 0.15, a proportion corresponding to less than 5 mosquitoes. To assess the inference uncertainty for distribution parameters (($\alpha$, $\beta$) and $\lambda$), it was more informative to evaluate the density distribution dispersion than the

**Table 1. Description of the intra-vector infection dynamic (IVD) model parameters inferred.**

| Parameter | Description | Interval of variation | Justification |
|---|---|---|---|
| $\gamma_I$ | Proportion of mosquitoes for which the infection barrier will be passed (= infection barrier) | [0,1] | Definition interval for proportional parameters |
| $\gamma_D$ | Proportion of mosquitoes for which the dissemination barrier will be passed (= dissemination barrier) | [0,1] | Definition interval for proportional parameters |
| $\gamma_T$ | Proportion of mosquitoes for which the transmission barrier will be passed (= transmission barrier) | [0,1] | Definition interval for proportional parameters |
| $(\alpha, \beta)$ | Beta law parameters | $\geq 1 \leq 100$ | To avoid infinite distribution in 0 and to cover a large range of possible shapes |
| $\lambda$ | Exponential law parameters | [0,100] | To cover a large range of possible shapes |

parameter dispersion. To assess the density distribution dispersion, we first made a visual appreciation. For the exponential distributions, it was relatively straightforward to visualize dispersion. However, the visual appreciation of the dispersion of the beta distributions was more challenging. Consequently, for each distribution of each scenario, we conducted a comparison of each density distribution using a Kolmogorov-Smirnov test, which was then corrected using the FDR (false discovery rate) method [78]. This was followed by the calculation of the percentage of statistically similar distributions (p-value >0.05) for each scenario.

## Supporting information

**S1 Fig. IVD conceptual diagram.**
(PDF)

**S1 Appendix. Vectorial-competence experimental data.** This appendix provides additional figures and tables on the experimental data used for IVD model inference.
(PDF)

**S2 Appendix. Figures of additional results.** This appendix provides additional figures showing the results obtained from datasets that used the partial EID model. Additionally, for each scenario, figures are provided that include the main selected distributions, as well as the main selected dynamics in the different IVD stages.
(PDF)

**S3 Appendix. Statistical analysis of results.** This appendix contains further tables presenting the statistical analysis of the inference results.
(PDF)

## Acknowledgments

We thank Anne Lehebel and Nadine Brisseau (BIOEPAR, Nantes) for their advice on the statistical analyses of the results as well as the members of the BUNYA (IVPC, Lyon) and RAPSODI (INRIA, Lille) teams for their insightful contributions to the discussions.

**Disclaimer:** Views and opinions expressed are however those of the author(s) only and do not necessarily reflect those of the European Union or the granting authority. Neither the European Union nor the granting authority can be held responsible for them.

## Author contributions

**Conceptualization:** Léa Loisel, Vincent Raquin, Maxime Ratinier, Pauline Ezanno, Gaël Beaunée.

**Formal analysis:** Léa Loisel, Pauline Ezanno, Gaël Beaunée.

**Funding acquisition:** Vincent Raquin, Maxime Ratinier, Pauline Ezanno, Gaël Beaunée.

**Investigation:** Léa Loisel, Pauline Ezanno, Gaël Beaunée.

**Methodology:** Léa Loisel, Pauline Ezanno, Gaël Beaunée.

**Software:** Léa Loisel, Pauline Ezanno, Gaël Beaunée.

**Visualization:** Léa Loisel, Pauline Ezanno, Gaël Beaunée.

**Writing – original draft:** Léa Loisel.

**Writing – review & editing:** Léa Loisel, Vincent Raquin, Maxime Ratinier, Pauline Ezanno, Gaël Beaunée.

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
