## [Decision Letter · Decision Letter 0]

19 Dec 2024

PCOMPBIOL-D-24-01822

Within-vector viral dynamics challenges how to model extrinsic incubation period for major arboviruses: dengue, Zika, and chikungunya

PLOS Computational Biology

Dear Dr. Loisel,

Thank you for submitting your manuscript to PLOS Computational Biology. After careful consideration, we feel that it has merit but does not fully meet PLOS Computational Biology's publication criteria as it currently stands. Therefore, we invite you to submit a revised version of the manuscript that addresses the points raised during the review process.

Please submit your revised manuscript within 60 days Feb 18 2025 11:59PM. If you will need more time than this to complete your revisions, please reply to this message or contact the journal office at ploscompbiol@plos.org. Please include the following items when submitting your revised manuscript:

We look forward to receiving your revised manuscript.

Kind regards,

Benjamin Althouse

Section Editor

PLOS Computational Biology

Benjamin Althouse

Section Editor

PLOS Computational Biology

**Journal Requirements:**

2) Please ensure that the funders and grant numbers match between the Financial Disclosure field and the Funding Information tab in your submission form. Note that the funders must be provided in the same order in both places as well. State the initials, alongside each funding source, of each author to receive each grant. For example: "This work was supported by the National Institutes of Health (####### to AM; ###### to CJ) and the National Science Foundation (###### to AM)."

**Reviewers' comments:**

Reviewer's Responses to Questions

**Comments to the Authors:**

Reviewer #1: Please see attached

Reviewer #2: Overall, I enjoyed reading this manuscript but there are some minor revisions I recommend.

Intro

- I find the purpose of this paper to be poorly explained, it is well-known that only a small fraction of arbovirus infections in mosquitoes end up being transmitted. There are decades of empirical data showing this, this paper tries to argue that this is a new finding. I think the authors need to be more careful in the phrasing they use to describe to purpose and results of this manuscript.

- During the introduction the authors refer to compartment often (line 86), please define this compartment.

- The reviewers need to note in their manuscript that EIP is not solely used to depict vector-borne disease transmission, it is typically accompanied by infection, dissemination, or transmission rates to calculate the R0 or the vectorial capacity.

Results

- This manuscript is currently formatted where the results come before the methods, if you intend to keep the manuscript formatted this way the authors should better explain the model in the results section (line 132).

- Describe what you mean by “scenario” in the results, methods, and figure legend. It would seem from your supplemental that it is data from one manuscript, but this needs to be well described in the main text.

- Fig 1 panel i is hard to understand, I recommend authors change the histogram pattern rather than just the direction of the lines and add a black line to denote a break in the bar

- Have the authors investigated how different strains, and populations (colony vs field) change their model?

Methods

- Why was such a small number of research articles chosen for the model? I would appreciate the authors outlining the reason they picked these specific manuscripts and not others in the methods of the manuscript rather than a graphical depiction in the supplemental. I can think of several other mosquitoes on DENV/ZIKV infection in mosquitoes that were not included, wouldn’t more scenarios help improve your model? (line 292)

- If the main point of this paper is the generic model then the model should be included in the main text of the MS. I recommend the authors move figure S2.2 to the main text and add more detail about the model to the figure

- Fig S2.2 is very confusing, what does the number of lines mean on the left side of the figure, please rewrite the figure legend.

Discussion

- What other abiotic factors should be tested (line 282)?

- A lot of mosquito studies have a sample size of 40-50 mosquitoes per treatment (line 328), 20 is the lowest you can go and is rarely used now, please correct this.

- One of the major caveats of this MS is that the authors state they only used open-access papers with data available. People rarely publish their raw data, but one can often infer from the tables in the manuscript, especially in vector biology manuscripts where the data is described in detail (n, n infected, n disseminated, n transmitting, and dpi) (line 259). Have the authors thought about running the model with non-open access manuscripts and does that change the result?

Reviewer #3: The manuscript submitted by Loisel et al describes a compartmental model to better predict the probability that exposed mosquitoes become infectious during the extrinsic incubation period for a variety of arboviruses. The authors use data found in existing literature to evaluate the ability of their compartmental model to predict experimentally-derived infection probabilities. While the model presented fits the experimental data well and provides a useful framework for modeling the probability of infectious mosquitoes for better understanding transmission, there is some confusion in how the proposed model fits in with existing frameworks used to model vector-borne disease transmission. For example, in most equations of vector-borne disease transmission such as R0 or vectorial capacity, in addition to EIP there is usually a vector competence variable to describe the probability of an exposed mosquito to become infectious. The authors don’t seem to emphasize how this has been done already in the introduction and discussion and make it seem like EIP is used independently of vector competence in many transmission models when this is not the case. I think the manuscript needs to be revised to better acknowledge how the authors’ newly presented model with three compartments in IVD compares to typical presentations of a single parameter for vector competence in the equation for R0 and vectorial capacity instead of only comparing to one exponential approach vs the author’s approach. Does the work presented here allow researchers to better estimate the vector competence variable already used in most transmission models? Or is it independently used to estimate vector competence for a different type of transmission model? Some clarity on how this new framework fits in with existing frameworks rather than just a single variable within other frameworks (EIP) is needed. For example R0 for vector-borne diseases typically is described as (N(ah)2bcpn)/H(-ln(p))r where N represents vector abundance, a the daily biting rate of the vector, h the proportion of bites by vectors on susceptible hosts, b represents the per bite probability of a vector becoming infected given exposure, c is the per bite probability of a host becoming infected given exposure, p is the vector daily survival rate, n is the extrinsic incubation period of the parasite in days, H is the abundance of susceptible hosts in the population, and r is the daily recovery rate of infected hosts. I understand that EIP could be a distribution and assumes that all mosquitoes have the same EIP but variation in mosquitoes becoming infectious is already accounted for elsewhere in the R0 equation by the variables p and b. It would be helpful if the authors could explain why the three compartments of IVD are more useful to have than a single compartment currently in use in most models. As the manuscript currently reads, there is more focus on the EIP which is a separate variable from vector competence and should be more clearly explained. Below, I’ve made some suggestions line by line for where I think additional improvements in this manuscript could be made. Overall, I think it is well written and the model is presented as a useful tool for better predicting vector transmission dynamics.

Major edits:

Lines 86 -87: There are examples where vector competence is not represented as the inverse of EIP, discuss those examples and how those examples perform compared to the framework you present here.

Lines 101 - 112: I think it might be useful to show the big picture transmission model most researchers use to represent transmission and which part of that model the work you present here impacts. For example, the authors call out the EIP parameter but I think the work here more directly predicts the vector competence parameter. Provide an explanation for how these variables relate and which part(s) of transmission models the authors are improving in this study (i.e. estimates of vector competence by using a 3 compartment model). Is it the EIP parameter or the vector competence parameter and how it varies across EIP?

Lines 260-1: “Although the existence of these barriers is already known (33), the majority of epidemiological models of vector-borne disease transmission does not take them into account and assumes that all exposed mosquitoes eventually become infectious” – in most models of R0 or vectorial capacity for vector-borne diseases, there is a vector competence term that is not assumed to be 1. For example, see Smith DL, McKenzie FE, Snow RW, Hay SI (2007) Revisiting the Basic Reproductive Number for Malaria and Its Implications for Malaria Control. PLOS Biology 5(3): e42. https://doi.org/10.1371/journal.pbio.0050042, where c is used to describe the probability that a mosquito becomes infected given a bite on an infectious human. There are many papers in the malaria context where transmission models contain parameters to describe the probability that a mosquito becomes infected and most assume that this probability is not 1 and varies due to the barriers already mentioned in your manuscript. There is also substantial literature which already acknowledges variation in transmission probabilities due to variability in the barriers or things that impact these barriers, for example see Lefevre T, Ohm J, Dabiré KR, et al. Transmission traits of malaria parasites within the mosquito: Genetic variation, phenotypic plasticity, and consequences for control. Evol Appl. 2018; 11: 456–469. https://doi.org/10.1111/eva.12571. Please acknowledge the approach others have taken to incorporate variation in vector competence in transmission models here and compare your method to methods previously used. In Aedes, there are equally plentiful examples of prior work which acknowledges variation in vector competence which is accounted for in the vectorial capacity equation, for example: Effect of Larval Competition on Extrinsic Incubation Period and Vectorial Capacity of Aedes albopictus for Dengue Virus Bara J, Rapti Z, Cáceres CE, Muturi EJ (2015) Effect of Larval Competition on Extrinsic Incubation Period and Vectorial Capacity of Aedes albopictus for Dengue Virus. PLOS ONE 10(5): e0126703. https://doi.org/10.1371/journal.pone.0126703.

Line 279: “considerable additional amount of data would be required” – what data is required? Provide recommendations for what type of data need to be experimentally collected and why to help readers understand the what and why of this statement. You mention some of these things later in the discussion, but would be good to list earlier.

Lines 281 - 284: Could the authors provide specific examples of the types of data and experiments that are currently lacking but would be informative to the field? Similarly, what would the proposed standardized data format be? What sorts of standards would the authors recommend?

Lines 330 - 334: It would be very useful to experimental biologists to be given more clear recommendations on things like sample sizes for experiments – I wonder if the authors could provide examples of experimental designs they would recommend for different types of experiments based on the model results and their experience using the existing data in the literature to support this recommendation

Lines 402 - 416: why is it better to compartmentalize IVD into 3 stages instead of just having one transmission probability variable?

Minor edits:

Abstract: Rephrase “To elucidate transmission, anticipate their spread and efficiently control them, mechanistic modelling has proven its usefulness.” for clarity. Suggestion: “Mechanistic modeling has proven useful to elucidate transmission, anticipate the spread, and predict the response of arboviruses to control tools.”

Abstract: Missing “on” in “most models rely assumptions” in cover page (but appears in abstract of main doc)

Abstract, line 23: change “two strong hypotheses” to “two strong assumptions”?

Abstract, line 25: change “To assess these hypotheses” to “To assess the validity of these assumptions”

Line 39: change “poses” to “posed”

Line 42: specify for arboviruses instead of just general epi models?

Line 90: change “every exposed mosquitoes” to “every exposed mosquito”

Line 91: change “is better represented by non-exponential distribution to “is better represented by a non-exponential distribution”

Line 105: change “infection” to “infections”

Figure 1: there are gray bars in A-C, ii and iii around x=day10, y=0.5. What are these designating?

Line 147: Figure 1 caption missing (i) description. Why do “modBetaBeta, modBetaExpo, modExpoBeta, modExpoExpo” listed in caption not match models listed in top legend (“IbetaDexpo, IbetaDbeta, IexpoDexpo, IexpoDbeta”)?

Line 154: change “exposition” to “exposure”

Line 232: change “demonstrating” to “demonstrated”

Line 245: change diseases to disease

Line 256: mention what is meant by “others”. Suggest “other vector-borne diseases like malaria”?

Line 262: changes “allows to” to “allows us to”

Line 313: change “defining” to “define”

Line 359 - 361: wording is unclear, please rephrase for clarity

Line 394: change “This could enable to study” to “This could enable researchers to study”

Figure S2.2: Change “on body”, “on saliva” to “in body”, “in saliva”

Figure 1: “effectif” is in French? Translate to English

**Have the authors made all data and (if applicable) computational code underlying the findings in their manuscript fully available?**

Reviewer #1: Yes

Reviewer #2: None

Reviewer #3: Yes

PLOS authors have the option to publish the peer review history of their article (what does this mean? ). If published, this will include your full peer review and any attached files.

**Do you want your identity to be public for this peer review?** For information about this choice, including consent withdrawal, please see our Privacy Policy .

Reviewer #1: No

Reviewer #2: No

Reviewer #3: No

**Figure resubmission:**
---

## [Decision Letter · Decision Letter 1]

6 Apr 2025

PCOMPBIOL-D-24-01822R1

Intra-vector infection dynamics challenges how to model extrinsic incubation period for major arboviruses: dengue, Zika, and chikungunya

PLOS Computational Biology

Dear Dr. Loisel,

Thank you for submitting your manuscript to PLOS Computational Biology. After careful consideration, we feel that it has merit but does not fully meet PLOS Computational Biology's publication criteria as it currently stands. Therefore, we invite you to submit a revised version of the manuscript that addresses the points raised during the review process.

Please submit your revised manuscript within 30 days Jun 06 2025 11:59PM. If you will need more time than this to complete your revisions, please reply to this message or contact the journal office at ploscompbiol@plos.org. Please include the following items when submitting your revised manuscript:

We look forward to receiving your revised manuscript.

Kind regards,

Benjamin Althouse

Section Editor

PLOS Computational Biology

**Reviewers' comments:**

Reviewer's Responses to Questions

**Comments to the Authors:**

**Please note that one of the reviews is uploaded as an attachment.**

Reviewer #1: Please see attached

Reviewer #2: The authors of this MS set out to model how EIP is not exponential but rather a beta distribution. As stated earlier I do feel that this paper could have analyzed more data sets and is missing some of the key knowledge already shared in the field of arbovirology. The authors could be more diligent in providing a fuller picture in the discussion section, outlining the impact this will have on the field and explaining their results more clearly, more specifically what addition it is bringing to the field. Overall, the authors have done a fine job addressing edits, but I wish to see several other changes made to this MS.

Major

- If the main purpose of this manuscript is to model how EIP is not exponentially increasing but rather unimodal then I would suggest changing the title of this MS. Additionally, as I stated in my last review I am not convinced that these findings are novel.

- Line 166 – when you say selected you mean the model selected it, I think a different word would be better suited here to describe this result.

- Line 295 -A caveat that should be mentioned is that crossing each of these barriers within the mosquito includes a bottleneck, thus the population of the virus is changing and populations that perform well in one compartment may not be well suited for the other. I would review the literature on quasispecies dynamics

https://www.sciencedirect.com/science/article/pii/S2589004223017881

https://www.cell.com/cell-reports/fulltext/S2211-1247(17)30453-9

https://www.sciencedirect.com/science/article/pii/S1879625716301365

https://www.mdpi.com/1999-4915/6/10/3991

- Line 398 – it’s common for the infection to go up after the initial timepoint, you should cite the studies where infection goes up (line 400)

- Line 423- I would also mention coinfections could influence the disease dynamics within the mosquito

- Line 436 – stating this could be used for transmission control is a stretch, how would a public health official utilize this model in their day-to-day work?

Minor

Line 165 – rather than saying infected 2x I would say taking an infectious bloodmeal

Line 270 – should state open access

Line 308 – R0 should be written as a subscript

Line 168 – I’m not sure why Fig2C is being discussed before Fig 2A but if this is how you want it formatted then you should change the order of the panels.

Reviewer #3: Thank you for responding and addressing reviewer comments. All my concerns have been addressed.

**Have the authors made all data and (if applicable) computational code underlying the findings in their manuscript fully available?**

Reviewer #1: Yes

Reviewer #2: Yes

Reviewer #3: Yes

PLOS authors have the option to publish the peer review history of their article (what does this mean? ). If published, this will include your full peer review and any attached files.

**Do you want your identity to be public for this peer review?** For information about this choice, including consent withdrawal, please see our Privacy Policy .

Reviewer #1: No

Reviewer #2: No

Reviewer #3: No

**Figure resubmission:**
---

## [Decision Letter · Decision Letter 2]

4 Aug 2025

Dear Miss Loisel,

We are pleased to inform you that your manuscript 'Intra-vector infection dynamics challenge how to model the extrinsic incubation period for major arboviruses: dengue, Zika, and chikungunya' has been provisionally accepted for publication in PLOS Computational Biology.

Best regards,

Benjamin Althouse

Section Editor

PLOS Computational Biology

Benjamin Althouse

Section Editor

PLOS Computational Biology

Reviewer's Responses to Questions

**Comments to the Authors:**

Reviewer #1: The manuscript is much improved compared to the previous version and I am satisfied with the responses provided from my previous comments. I still think that some areas can be tightened up with respect to descriptions and language in the Results and Discussion, but this wouldn't prevent publication. Just ensure you have a final read through and try and make statements as clear and concise as possible.

Reviewer #2: I am pleased with the revisions by the authors and have no further comments.

**Have the authors made all data and (if applicable) computational code underlying the findings in their manuscript fully available?**

Reviewer #1: Yes

Reviewer #2: Yes

PLOS authors have the option to publish the peer review history of their article (what does this mean? ). If published, this will include your full peer review and any attached files.

**Do you want your identity to be public for this peer review?** For information about this choice, including consent withdrawal, please see our Privacy Policy .

Reviewer #1: No

Reviewer #2: No

---

## [Editor Report · Acceptance letter]

PCOMPBIOL-D-24-01822R2

Intra-vector infection dynamics challenge how to model the extrinsic incubation period for major arboviruses: dengue, Zika, and chikungunya

Dear Dr Loisel,

I am pleased to inform you that your manuscript has been formally accepted for publication in PLOS Computational Biology. Your manuscript is now with our production department and you will be notified of the publication date in due course.

With kind regards,

Lilla Horvath
